Pulmonary transcriptomic responses indicate a dual role of inflammation in pneumonia development and viral clearance during 2009 pandemic influenza infection

Almansa Raquel 1
Martínez-Orellana Pamela 2
Rico Lucía 1
Iglesias Verónica 1
Ortega Alicia 1
Vidaña Beatriz 3
http://orcid.org/0000-0001-5246-7276 Martínez Jorge 2 4
Expósito Ana 1
http://orcid.org/0000-0002-5703-7360 Montoya María 2 5
Bermejo-Martin Jesús F. 1 jfbermejo@saludcastillayleon.es
1 Laboratory of Biomedical Research in Sepsis (BIOSEPSIS), Hospital Clínico Universitario de Valladolid, Instituto de Estudios de Ciencias de la Salud de Castilla y León (IECSCYL) , Valladolid , Spain
2 Centre de Recerca en Sanitat Animal (CReSA), Universitat Autónoma de Barcelona, IRTA , Barcelona , Spain
3 Department of Pathology, Animal and Plant Health Agency (APHA) , Surrey , UK
4 Departament de Sanitat i d’Anatomia Animals, Universitat Autónoma de Barcelona , Barcelona , Spain
5 African Swine Fever Virus Immunology Group, The Pirbright Institute , Surrey , UK
Roberts Craig
Electronic publication date: 2017 Oct 11
Publication date: 2017
Volume: 5
Electronic Location ID: e3915
Received 2017 May 23; Accepted 2017 Sep 21
Copyright: © 2017 Almansa et al.
Copyright year: 2017
Copyright holder: Almansa et al.
License: This is an open access article distributed under the terms of the Creative Commons Attribution License, which permits unrestricted use, distribution, reproduction and adaptation in any medium and for any purpose provided that it is properly attributed. For attribution, the original author(s), title, publication source (PeerJ) and either DOI or URL of the article must be cited.
License URL: https://creativecommons.org/licenses/by/4.0/

Keywords: Influenza, Mice model, Inflammation, Gene expression, Immune response, Lung

Funding: Instituto de Salud Carlos III: Programa de Investigación Comisionada en Gripe GR09/0021 Programa para favorecer la incorporación de grupos de investigación en las Instituciones del Sistema Nacional de Salud EMER07/050 Biotechnology and Biological Sciences Research Council (BBSRC) BBS/E/I/00002014 This work was supported by Instituto de Salud Carlos III: “Programa de Investigación Comisionada en Gripe (GR09/0021) and Programa para favorecer la incorporación de grupos de investigación en las Instituciones del Sistema Nacional de Salud” (EMER07/050). This work was also supported by Biotechnology and Biological Sciences Research Council (BBSRC) grant (BBS/E/I/00002014). The funders had no role in study design, data collection and analysis, decision to publish, or preparation of the manuscript.

==============================
Background

The interaction between influenza virus and the host response to infection clearly plays an important role in determining the outcome of infection. While much is known on the participation of inflammation on the pathogenesis of severe A (H1N1) pandemic 09-influenza virus, its role in the course of non-fatal pneumonia has not been fully addressed.

Methods

A systems biology approach was used to define gene expression profiles, histology and viral dynamics in the lungs of healthy immune-competent mice with pneumonia caused by a human influenza A (H1N1) pdm09 virus, which successfully resolved the infection.

Results

Viral infection activated a marked pro-inflammatory response at the lung level paralleling the emergence of histological changes. Cellular immune response and cytokine signaling were the two signaling pathway categories more representative of our analysis. This transcriptome response was associated to viral clearance, and its resolution was accompanied by resolution of histopathology.

Discussion

These findings suggest a dual role of pulmonary inflammation in viral clearance and development of pneumonia during non-fatal infection caused by the 2009 pandemic influenza virus. Understanding the dynamics of the host’s transcriptomic and virological changes over the course of the infection caused by A (H1N1) pdm09 virus may help identifying the immune response profiles associated with an effective response against influenza virus.

Introduction

Influenza is one of the most common respiratory infectious diseases and a worldwide public health concern. The World Health Organization (WHO) estimates that influenza viruses infect around 5–15% of the global population, resulting into 250,000 to 500,000 deaths each year (Vemula et al., 2016).

At the beginning of 2009, a new influenza virus of the subtype H1N1, [A (H1N1) pmd09], was detected in Mexico. The vast majority of infections caused by this new strain were mild and self-limiting upper respiratory tract illness. However, a small percentage of patients infected by the A H1N1 pm09 virus developed primary viral pneumonia, resulting in respiratory failure, acute respiratory distress, multi-organ failure and death (Health Protection Agency et al., 2009). A large proportion of these severe cases occurred in young adults with accompanying co-morbidities (chronic respiratory disease, cardiovascular disease, hypertension, obesity and diabetes) (Jain et al., 2009).

The host response to the infection clearly plays an important role in determining the outcome of the patients infected by influenza viruses (Almansa, Bermejo-Martín & de Lejarazu Leonardo, 2012). In this regard, severely infected patients by the influenza A (H1N1) pdm09 virus was characterized by the presence of high plasmatic levels of cytokines, chemokines and other immune mediators accompanying the presence of pneumonic infiltrates (Bermejo-Martin et al., 2009; Hagau et al., 2010; To et al., 2010). Moreover, we have shown that systemic levels of these mediators were directly associated with viral levels secreted by the respiratory tract from the beginning of the disease (Almansa et al., 2011a). In addition, persistence of viral secretion has been found in the patients with the worst outcomes (Lee et al., 2009), paralleling the presence of impaired expression of a number of genes participating in adaptive immune responses. Depression of adaptive immunity response has been previously correlated with poor control of infection and maintenance of inflammation, and secondarily with the generation of damage to the infected tissues with the development of further respiratory failure (Bermejo-Martin et al., 2010).

While much is known about the immune alterations and the participation of inflammation on the pathogenesis of severe A (H1N1) pandemic influenza, their role in the course of non-fatal pneumonia has not been sufficiently studied. Aimed to clarify this role, we employed a systems biology approach to study gene expression profiles (GEP) and its relation to histology and viral dynamics in the lungs of healthy immune-competent mice with pneumonia caused by human influenza A (H1N1) pdm09 virus, which successfully resolved the infection.

Materials and Methods

Ethics statement

The ethical protocol and the research were reviewed and approved by the Animal and Human Experimentation Ethical Committee of the Autonomous University of Barcelona (Internal Register Number: 1124M2R) and the Ethical Animal Experimentation Commission of the Catalan Government (Register Number: 5767).

All the animal experiments were done at the Biosafety level 3 (BSL3) facilities of the Centre de Recerca en Sanitat Animal (CReSA, Barcelona, Spain). Animal care was performed according to the standard procedures of the center (Martínez-Orellana et al., 2015). Seven-week-old C57BL6/JOlaHsd (C57BL6) female mice (Harlan Laboratories, Barcelona, Spain) were housed in groups in experimental isolation cages for one week in acclimation (72 animals in total). Throughout the experiment, all mice were provided with commercial food pellets and tap water ad libitum.

A (H1N1) pdm 2009 Catalonian virus and mice infection

A human pandemic influenza A virus, A/Catalonia/63/2009 (CAT09) (GenBank accession numbers GQ464405–GQ464411 and GQ168897) was used for animal infection (Busquets et al., 2010). CAT09 was passaged in MDCK two times and the viral stock had a titer of 106 PFU/ml. Animals were divided into two groups of 32 mice each; distribution was done as follows: untreated control group (mock group) and pdmH1N1 2009 infected-group (CAT09). To evaluate the pathogenicity mice were infected through intranasal instillation with 50 μl CAT09 at 104 PFU/mice as described previously (Itoh et al., 2009). Successful CAT09 mice infection and pathogenicity was previously confirmed by our experimental work (Orellana-Martínez, 2014). Control non-infected mice were treated with 50 μl phosphate-buffered saline (mock infection) to reproduce CAT09 infection.

Mice monitoring and sampling

For 10 days, mice were observed daily to record changes in body weight and clinical signs. Based on our previous experimental work, the day showing the most important histological changes in the lung following infection caused by CAT09 is day 5, while resolution of histological changes occurs by day 10. Consistent with our previous experience (Orellana-Martínez, 2014), necropsies of 12 animals per group were performed at days 1, 5 and 10 post infection (dpi). Animals were euthanized with intraperitoneal inoculation of penthobarbital under anesthesia with 5% isofluprane and tissue samples of lung were dissected from dead animals using the standard surgical procedures. Lung samples of six animals per group were used for viral load determination and histological examination. Lung samples were snap-frozen on dry ice and stored at −80 °C until further processing. Gene expression profiling was performed for whole lungs of the other six animals per group by using microarrays.

Determination of viral load

Viral quantification was determined by plaque assay determining plaque-forming units (PFU) following our laboratory standard operating procedures (Martínez-Orellana et al., 2015). Briefly, supernatants were obtained after weighing, homogenizing and centrifuging lung samples. 0.1 ml of 10-fold supernatant dilutions were incubated with MDCK cells plated in 12-well tissue cultures plates for 1 h. Then, cells were washed with phosphate buffer saline and plates were overlaid with 1.4% noble agar (Becton Dickinson, Pont-de-Claix, France), mixed 1:1 with 0.5 μg/ml of bovine trypsin and minimum essential medium eagle (MEM) (both from Sigma-Aldrich SA, Madrid, Spain) supplemented with 100 UI/ml penicillin and 100 μg/ml streptomycin (Invitrogen®, Barcelona, Spain). After four days of incubation, cells were fixed for 20 min using 10% formalin (Sigma-Aldrich SA, Madrid, Spain) and then overlaid with 1% crystal violet (Anorsa, Barcelona, Spain). Finally, cells were washed with water in order to visualized plaques, which were counted and compared to uninfected cells.

Histopathology

Lung samples were collected for macroscopical and histological examination according to our laboratory standard operating procedures (Martínez-Orellana et al., 2015). The procedures involved lung sample fixation using neutral-buffered 10% formalin for 48 h, followed by embedment in paraffin wax. Next, sections of 3 μm were stained using haematoxylin and eosin (HE). Cross sections of the lungs were analyzed separately. A semi-quantitative assessment of IAV-associated microscopic lesions in the lungs was performed for each animal. The lesional scoring was graded on the basis of lesion severity as previously described by Vidaña et al. (2014).

RNA extraction and microarray processing and analyzing

At designated time points (1, 5 and 10 dpi), C57BL6 mice were euthanized and lung tissue was collected in RNAlater and stored at −80 °C until further processing. Total RNA was extracted from lung samples using the Ribopure kit (Ambion, Life technology, Carlsbad, CA, USA). RNA integrity and concentration were evaluated as previously described (Almansa et al., 2015). A total amount of 100 ng of mRNA was processed as described to obtain Cyanine 3-CTP-labeled cRNA (Almansa et al., 2015). Next cRNA was hybridized with Mouse GE 4 × 44K v2 Microarray Kit (Agilent p/n G4846A) overnight (17 h) at 65 °C on a rotator. Image acquisition was performed using an Agilent Microarray Scanner (Agilent G2565CA, Santa Clara, CA, USA) and data were extracted using the Agilent Feature Extraction Software 10.7.1.1 following the Agilent protocol GE1-107_Sep09. Raw data were collected and preprocessed by using the GeneSpring GX 12.0 software (Almansa et al., 2015). This software was employed also to perform the statistical analysis, which involved the use of a moderate T-test to identify those genes showing significant differences between their expression levels fixing a p < 0.05 with further application of the Benjamini–Hochberg correction for multiple comparisons. A fold change in gene expression ≥2 was used to obtain the list of those genes showing the more important variations in their expression levels between groups along time (1, 5 and 10 dpi). Ingenuity pathway analysis (IPA) (Ingenuity Systems-Quiagen, Redwood City, CA, USA) was employed to determine whether a canonical pathway is enriched with genes of interest by using Fisher’s exact test.

Microarray data accession number

Microarray expression data sets were uploaded at the Array Express microarray data repository and are available publicly under accession number E-MTAB-3866.

Validation of gene expression results from microarrays

Results of gene expression obtained using microarrays were confirmed by using a next generation PCR technology, droplet digital PCR (ddPCR), using the Bio-Rad QX200™ Droplet Digital™ PCR system. About 5 ng of total mRNA were retro-transcribed to cDNA and analyzed by ddPCR using a Bio-Rad QX200™ platform as previously described (Tamayo et al., 2014). Quantification of expression levels of target mRNAs was performed using pre-designed TaqMan® Assay Primer/Probe Sets, (FAM-labeled MGB probes, Thermo Fisher/Scientific-Life Technologies, Waltham, MA, USA): IL6 gene; interleukin 6 (Reference: Mm00446190_m1) and IFNB1 gene; interferon beta 1 (Reference: Mm00439552_s1). The droplet reader used at least 10,000 droplets to determine the percentage of positive droplets and calculation of copy number of cDNA per ng of initial mRNA. Spearman correlation between ddPRC and microarrays results was performed using SPSS 15.0 (Fig. S1).

Statistical analysis

SPSS 15.0 software was employed for perform statistical comparison of weight loss and viral load between groups at all sampling times (SPSS Inc., Chicago, IL, USA). The statistical test used was the U Mann–Whitney, and the significance level (α) was set at 0.05. All graphs used for represent the variations on weight loss and viral load were performed using GraphPad Prism 6 (GraphPad Software, La Jolla, CA, USA).

Results

A (H1N1) pdm09 virus infection induced moderate weight loss during the first five days of infection

Weight was evaluated each day during the first 10 days following infection with the pandemic CAT09 virus. Even though the percentage of body weight loss in CAT09-infected animals was not dramatic, CAT09-infected mice showed significantly greater weight loss on the first five days compared to uninfected controls (p < 0.05). After 5 dpi, infected mice began to recover their normal weight with no significant differences compared to mock mice (Fig. 1A).

Figure 1 Changes in body weight and lung viral load induced by A (H1N1) pdm09 virus.

(A) Average weight curve for C57BL6 mice infected through intranasal instillation with 50 μL CAT09 at104 PFU A/Catalonia/63/2009 (H1N1pdm) and mock. (B) Viral load in lung homogenates collected at days 1, 5 and 10 pi (n = 6 for all groups). Infection of Madin–Darby Canine Kidney cells was employed to measure viral titers. The U Mann–Whitney test was used to compare weight loss and viral load between groups at all sampling time. The significance level (α) was set at 0.05. Asterisks indicate significant differences between groups (A) or between times points (B).

Human A (H1N1) pdm09 virus causes a productive infection in the lower respiratory tract of mice

Virus titers in lung homogenates measured on 1, 5 and 10 dpi are shown in Fig. 1B (n = 6 mice per group). The highest value in viral load detected was one day after infection (average: 1.08E + 05 PFU/g, SD: 1.43E + 05). However, day 5 pi, infected animals were still secreting virus in lungs [1.01E + 04 PFU/g, 0.86E + 04], becoming undetectable at day 10 pi.

CAT09-infected mice developed pneumonia at day 5 post-infection, fully recovering at day 10 post-infection

Lung tissues from six animals per group were histopathologically examined at day 1, 5 and 10 pi. As expected, control animals showed no histopathological lesions (Fig. 2). Microscopic lesional scores were assigned for each animal (Fig. 2B). At 1 dpi, three of six infected mice presented histopathological lesions, two of them exhibited necrotizing bronchiolitis and the other one presented bronchointerstitial pneumonia. At day 5 pi, five of six animals presented severe bronchointerstitial pneumonia consisting of moderate to high numbers of lymphoplasmacytic cells and neutrophils infiltrated the bronchiole and surrounding alveoli (Fig. 2). Nevertheless, day 10 pi was characterized by the total resolution of lung lesions in the CAT09-infected animals.

Figure 2 Histopathology of mice belonging to control and CAT09 groups at day 1, 5 and 10 pi.

(A) Hematoxilin/Eosin stain. Arrows indicate the infiltrate in the viral infected lungs. (B) Microscopic lesional scores: grade 0 (no histopathological lesions observed), grade 1 (mild to moderate necrotizing bronchiolitis), grade 2 (bronchointerstitial pneumonia characterized by necrotizing bronchiolitis and diffuse alveolar damage in adjacent alveoli), and grade 3 (necrotizing bronchiolitis and diffuse alveolar damage in the majority of the pulmonary parenchyma) (Vidaña et al., 2014).

A (H1N1) pdm09 virus induced changes in gene expression levels in the lungs

Gene expression profiles at lungs were compared between six infected animals and six mock mice at days 1, 5 and 10 pi. No differences in GEP were found at day 1 pi (Fig. 3A), but important differences were observed at day 5 pi, paralleling the development of histological pneumonia (Fig. 3B; Table S1). In the CAT09-infected mice group, 1,264 genes showed a significant variation of their expression levels by day 5 pi compared to the control group (418 upregulated and 847 down expressed) (Fig. 3B; Table S1). Genes showing the most important differences between both groups were interleukin 6 (IL6) (Fold change FC: 86.6), interferon beta 1 (IFNb) (FC: 62.6) and chemokine (C–X–C motif) ligand 10 (IP10), (FC: 43.3) (Figs. 3E and 3F; Table S1). Expression levels of the vast majority of genes normalized by day 10, coinciding with virus clearance and resolution of histological changes (Figs. 3C, 3E and 3F; Table S1). Only 30 out of the 1,264 genes kept on showing altered expression levels by day 10 pi (Fig. S2; Table S1). Interestingly, expression levels of IL6 persisted remarkably high by this time point (FC: 10.91) along with those of granzyme K (Gzmk) (FC: 15.8) (Fig. S2).

Figure 3 Pulmonary gene expression profiles at day 1, 5 and 10 post infection.

(A–C) Volcano plots for the representation of the number of genes with significant variation of their expression levels between CAT09 and mock groups, at different time points (1 (A), 5 (B) and 10 (C) dpi). The level of significance was fixed in p < 0.05, with Benjamini–Hochberg multiple testing corrections and Fold change >2. The list of genes differentially expressed between groups is shown in Table S1. (D) Top 20 Canonical signaling pathways altered by A (H1N1) pdm09 virus. The x-axis represents the percent of genes of each pathway whose expression levels were altered by the virus. Genes involved in the top 20 canonical signaling pathways are shown in Table S2. (E and F) Gene expression levels of cytokines, chemokines: (E) and IFN-stimulated genes (F) during infection with A (H1N1) pdm09 influenza virus. The heat map depicts the most representative immune response-related genes (yellow and blue colored genes in Table S3) that were differentially expressed between infection conditions at different time points. Colors represent the average value of gene expression levels of infected animals for each time point.

A (H1N1) pdm09 infection turned on the expression of genes involved in the innate response and in the switch to adaptive immunity by day 5 pi

Since most differences in gene expression were found by day 5 pi, we focused the ingenuity pathway analysis (IPA) on that day. The list of 1,264 genes (either up- or downregulated) was analyzed by IPA in order to identify the canonical pathways that were enriched at day 5 pi. Notably, cellular immune response and cytokine signaling were the two signaling pathway categories more representative of our analysis (Table S2). The most significant canonical pathways identified by IPA are described in Table 1 and Fig. 3D.

Table 1 Top 20 canonical signaling pathways altered by A (H1N1) pdm09 virus.

Ingenuity canonical pathways	p Value	Ratio	Top functions and diseases	
Role of hypercytokinemia/hyperchemokinemia in the pathogenesis of influenza	<0.001	0.244	Cell-to-cell signaling and interaction; cellular movement; hematological system development and function	
Hepatic fibrosis/hepatic stellate cell activation	<0.001	0.122	Organismal injury and abnormalities; cardiovascular system development and function; organismal development	
Communication between innate and adaptive immune cells	<0.001	0.165	Cell-to-cell signaling and interaction; cellular growth and proliferation; hematological system development and function	
Wnt/β-catenin signaling	<0.001	0.124	Gene expression; cellular development; tissue development	
Agranulocyte adhesion and diapedesis	<0.001	0.116	Cell-to-cell signaling and interaction; tissue development; hematological system development and function	
TREM1 signaling	<0.001	0.173	Cell-to-cell signaling and interaction; hematological system development and function; immune cell trafficking	
Differential regulation of cytokine production in intestinal epithelial cells by IL-17A and IL-17F	<0.001	0.304	Cell-to-cell signaling and interaction; hematological system development and function; immune cell trafficking	
Granulocyte adhesion and diapedesis	<0.001	0.113	Cell-to-cell signaling and interaction; hematological system development and function; immune cell trafficking	
Altered T cell and B cell signaling in rheumatoid arthritis	<0.001	0.148	Hematological system development and function; tissue morphology; cellular development	
Differential regulation of cytokine production in macrophages and T helper cells by IL-17A and IL-17F	<0.001	0.333	Cell-to-cell signaling and interaction; hematological system development and function; immune cell trafficking	
Role of IL-17F in allergic inflammatory airway diseases	<0.001	0.205	Connective tissue disorders; immunological disease; inflammatory disease	
Crosstalk between dendritic cells and natural killer cells	<0.001	0.146	Cell-to-cell signaling and interaction; cellular growth and proliferation; hematological system development and function	
HMGB1 signaling	<0.001	0.125	Cell-to-cell signaling and interaction; cellular movement; hematological system development and function	
Graft-versus-host disease signaling	<0.001	0.188	Cellular immune response; disease-specific pathways	
T helper cell differentiation	<0.001	0.155	Cell-mediated immune response; cellular development; cellular function and maintenance	
Atherosclerosis signaling	<0.001	0.122	Cell-to-cell signaling and interaction; cellular movement; hematological system development and function	
Role of macrophages, fibroblasts and endothelial cells in rheumatoid arthritis	<0.001	0.087	Cell death and survival; cellular development; cellular growth and proliferation	
Colorectal cancer metastasis signaling	<0.001	0.093	Cell death and survival; cell cycle; cellular development	
Role of osteoblasts, osteoclasts and chondrocytes in rheumatoid arthritis	0.001	0.091	Hematological system development and function; tissue morphology; cellular development	
Role of pattern recognition receptors in recognition of bacteria and viruses	0.001	0.110	Antimicrobial response; inflammatory response; infectious disease	
Note:

This table summarized the most significant canonical pathways identify by “ingenuity pathway analysis (IPA).” The IPA system implements Fisher’s exact test to determine whether a canonical pathway is enriched with genes of interest (the level of significance was fixed in p < 0.05). The ratio show the number of genes whose expression levels were different between CAT09 and mock groups, of the total of genes that have been described previously in each pathway.

Most of these pathways were involved in the innate immune response and inflammation: [Role of hypercytokinemia/hyperchemokinemia in the pathogenesis of influenza (Fig. 4), hepatic fibrosis/hepatic stellate cell activation, agranulocyte adhesion and diapedesis, TREM1 signaling, differential regulation of cytokine production in intestinal epithelial cells by IL-17A and IL-17F, granulocyte adhesion and diapedesis, altered T cell and B cell signaling in rheumatoid arthritis, differential regulation of cytokine production in macrophages and T helper cells by IL-17A and IL-17F, role of IL-17F in allergic inflammatory airway diseases, graft-versus-host disease signaling, role of macrophages, fibroblasts and endothelial cells in rheumatoid arthritis, role of pattern recognition receptors in recognition of bacteria and viruses and Wnt/β-catenin signaling pathway]. The vast majority of the genes involved in these pathways coded for cytokines (Table S2). H1N1 virus also induced alterations in pathways participating in the switch from innate to adaptive immunity: [Communication between innate and adaptive immune cells, crosstalk between dendritic cells and natural killer cells, T helper cell differentiation]. Table S3; Figs. 3E and 3F show the variation of the genes participating in these pathways along the study course.

Figure 4 Role of Hypercytokinemia/hyperchemokinemia in the pathogenesis of influenza signaling pathway.

“Ingenuity pathway analysis” identified this route as the most altered pathway of the analysis. Red: genes upregulated in the infected group compared with non-infected mice.

Discussion

The overarching aim of this work was to study the role of inflammation at pulmonary level during a non-fatal infection caused by the 2009 pandemic influenza virus using the mice model. In this sense, we analyzed the GEP and its relation to histology and viral dynamics in the lungs of healthy immune-competent mice with pneumonia caused by human influenza A (H1N1) pdm09 virus.

Our GEP analysis allowed us to identify the presence of marked activation of innate immunity genes by day 5 post infection, paralleling the existence of extensive pneumonic/cellular infiltrates in the lung, and active viral replication. The innate immune response is the first line of defense against invading viruses (Iwasaki & Pillai, 2014). Infection of the respiratory tract induced thus a typical antiviral response characterized by the activation of pro-inflammatory cytokines and interferon (IFNs) response genes (ISGs). In our analysis, the genes showing higher differences for their expression levels between infected mice and controls were IL6, IFNb, and IP10. These molecules, along with TNF and IL1b (also over-expressed at day 5), are the major cytokines limiting viral replication during influenza infection, recruiting immune cells to the sites of infection and producing inflammation (Nicholls, 2013).

IL6 is a pro-inflammatory cytokine which role in the pathogenesis of the A (H1N1) pdm09 remains unclear. There is a consensus in the literature about the existence of high systemic levels of IL6 in severe patients infected by A (H1N1) pdm09 virus (Bermejo-Martin et al., 2009; To et al., 2010; Zúñiga et al., 2011). This molecule induces pro-inflammatory responses such as leukocyte recruitment into the lung. Excessive production of IL6 has been associated with several pathological manifestations (Ho, Luo & Lai, 2015; Baillet et al., 2015). However, Paquette et al. (2012) demonstrated in IL6 deficient mice infected with A (H1N1) pdm09, that no significant differences in survival, weight loss, viral load, or pathology were observed between IL6 deficient and wild-type mice following infection. Based in our results, presence of high expression levels of this cytokine in the lung at day 10 could indicate that this cytokine plays a role in viral clearance and tissue repair after pneumonia. Other mouse models support the idea of a protective role of IL6 in influenza infections (Lauder et al., 2013).

IFNb is a cytokine member of type I interferon family. It induces an antiviral state in infected and neighboring cells (Ramos & Fernandez-Sesma, 2015). To do so, IFNs induce the transcription of hundreds of ISGs, which leads to numerous changes in the transcriptome of the cell. Interestingly, in our analysis, some OAS genes (OAS1a, OAS1f, OASL1 and OAS2), IFIT genes (IFIT1, IFIT2 and IFIT3), MX1, SOCS1 and CXCL10, all of them IGS genes, showed high expression levels in the infected mice compared with controls. The antiviral interferon response increased at day 5 pi, but decreased at day 10 pi, coinciding with viral clearance. These results are similar to the observations reported in previous studies using ferrets infected by A (H1N1) pdm09 virus (León et al., 2013; Rowe et al., 2010b). This authors showed an early robust innate ISG and chemokine response that shut down on days 7–10 pi, when viral load was undetectable.

The activation of a group of genes involved in the “Role of hypercytokinemia/hyperchemokinemia in the pathogenesis of influenza” pathway evidence the existence of a local “cytokine storm” in the lung, following infection by A (H1N1) pdm09 virus. Hypercytokinemia/hyperchemokinemia is a common finding that characterized an influenza infection at transcriptomic level (Morrison et al., 2014; Ma et al., 2011; León et al., 2013; Rowe et al., 2010b). Several experimental studies suggested that cytokine storm correlated directly with tissue injury and an unfavorable prognosis of severe influenza (Liu, Zhou & Yang, 2016). In our study, concomitant with high expression levels of IL6, IFNb and ISGs, the virus activated Th1 and chemokine responses mediated by IL1a, IL1b, IL12b, TNF, MCP1 and RANTES. These results are similar to those found at systemic level in patients with primary viral pneumonia (Bermejo-Martin et al., 2009; Hagau et al., 2010; To et al., 2010). In our model, the marked inflammatory program observed by day 5 in the lung got deactivated by day 10, paralleling resolution of histological changes and viral replication. Moreover, evaluation of gene expression levels along time in the infected group confirmed the appearance of a strong pro-inflammatory response at day 5 that is downmodulated at day 10 (Figs. 3E and 3F; Table S3). Similar results were also found in other experimental studies (Josset et al., 2012b; León et al., 2013; Rowe et al., 2010b), where the decreased of cytokine expression levels characterized the recovery phase of the disease. Therefore, a failure to effectively regulation of excessive inflammation may be, in part, responsible for severe cases of 2009-H1N1.

In turn, the activation of genes involved in “Agranulocyte adhesion and diapedesis,” “TREM1 signaling,” “Granulocyte adhesion and diapedesis,” “Graft-versus-host disease signaling” and “Role of pattern recognition receptors in recognition of bacteria and viruses” might confirm the existence of a transcriptomic program aimed to recruit lymphocytes, monocytes and neutrophils to the site of infection. Histological studies at day 5 pi confirmed the presence of extensive pneumonic/cellular infiltrates into the lung. Although the primary role of the innate immune response is limiting viral replication, excessive activation of innate immunity could induce tissue damage (Vidaña et al., 2014; de Jong et al., 2006). This phenomenon seems to occur also in the context of autoimmunity diseases such as rheumatoid arthritis (Catrina et al., 2016). In fact, “Altered T cell and B cell signaling in rheumatoid arthritis” and “Role of macrophages, fibroblasts and endothelial cells in rheumatoid arthritis” are two of the significant pathways identified by IPA in our analysis. In influenza disease, an exaggerated inflammatory response has been cited as the cause of pulmonary edema, alveolar hemorrhage and acute respiratory distress syndrome, conditions associated with necrosis and tissue destruction (To et al., 2001). Most of the genes participating in these pathways decreased their expression levels on day 10 pi paralleling resolution of pneumonia, reinforcing the idea that a correct modulation of inflammatory response is essential for recovery in this disease.

Ingenuity pathway analysis identified also three pathways related to interleukin 17: “Differential regulation of cytokine production in intestinal epithelial cells by IL-17A and IL-17F,” “Differential regulation of cytokine production in macrophages and T helper cells by IL-17A and IL-17F” and “Role of IL-17F in allergic inflammatory airway diseases.” Th-17 immunity participates in clearing pathogens during host defense reactions but is involved also in tissue inflammation in several autoimmune diseases, allergic diseases, and asthma (Nalbandian, Crispín & Tsokos, 2009; Cheung, Wong & Lam, 2008). In severe influenza it has been proposed to play a beneficial role (Iwakura et al., 2008; Bermejo-Martin et al., 2009; Almansa et al., 2011b).

Ingenuity pathway analysis also identified low expression levels of a group of genes involved in the “Wnt/β-catenin signaling pathway” at 5 pi. It has been previously described that influenza virus downregulates the expression of proteins of this pathway like FZD (Shapira et al., 2009). This is consistent with low expression levels of FZD2 and FZD7 genes found in our analysis. The biological repercussion associated to downmodulation of this pathway remains to be elucidated.

Finally, the activation of those cytokine genes involved in the [Communication between innate and adaptive immune cells, crosstalk between dendritic cells and natural killer cells, T helper cell differentiation] signaling pathways at day 5 pi could be reflecting the development of the adaptive immune response against the virus. Later activation of the adaptive immune response was previously supported by increased levels of granzyme mRNAs in blood cells (Rowe et al., 2010a). In our study, the virus induced the expression of granzyme A, B and K at day 5 pi. Moreover, expression levels of gramzyme K persisted remarkably high at day 10 pi, which is consistent with the data published in ferret infected by A (H1N1) pdm09 virus (Rowe et al., 2010a).

There are several works evaluating host transcriptomic responses to A (H1N1) pdm09 virus using animal models (Powell & Waters, 2017), with different scope, but not of all them properly integrate the GEP induced by the infection with the pathogenic events to built a comprehensive model to improve our understanding on the events underlying the appearance and resolution of pneumonia caused by influenza (see Table S4). The vast majority of these experimental models have focused on studying the host immune responses to the virus only on the acute phase of infection (Ma et al., 2011; Camp et al., 2012; Josset et al., 2012a; Zou et al., 2013; Morrison et al., 2014). The present work studies the relationship between host transcriptomic responses and the progression and resolution of infection caused by the A (H1N1) pdm09 influenza virus. The most similar study to the one we present here is that published by Rowe et al. (2010b). In that paper, the authors employ ferrets, which is one of the best models to reproduce human pathology in the context of influenza, but at the same time it is not an easily available model, being expensive and complicated to manage. Our study employed a mouse model, which is a more affordable, but nonetheless reproduced the major findings of Rowe et al., who evidenced the existence of a exuberant cytokine and chemokines response at the lungs paralleling histological changes, which was downmodulated following resolution of these changes (Rowe et al., 2010b). Our results support thus the potential use of this mice model for the study of immunopathology in influenza infection and for those works evaluating immunomodulators for the treatment of this disease. While other studies uses a mice adapted influenza strain such as PR8 (Pommerenke et al., 2012) or other mouse passaged 2009 strains, (Josset et al., 2012a; Manchanda et al., 2016), our study employs a strain obtained directly form a human patient. Mouse adaptation results in increased virulence and lung pathology and also induces a strong host transcriptional response after infection compared with non-adapted influenza strains (Josset et al., 2012a). In our opinion, our model could help to better understand the immune-pathogenic events on the basis of the most common scenario during the pandemics, which was that corresponding to a non-severe infection.

Conclusion

In conclusion, our findings suggest a dual role of pulmonary inflammation during non-fatal infection caused by the 2009 pandemic influenza virus. On one side, the activation in the lung of a marked innate immunity transcriptomic program was associated to the appearance of pneumonia, but on the other hand, activation of this program paralleled viral clearance (Fig. 5). Understanding the dynamics of the host’s transcriptomic and virus changes over the course of the infection caused by A (H1N1) pdm09 might help to identify the immune response profiles associated to effective/balanced responses against influenza virus.

Figure 5 Model of uncomplicated A (H1N1) pdm09 viral infection.

The virus induced the activation of a marked pro-inflammatory program at the lung level paralleling the emergence of histological changes. This program was associated to viral clearance, and its resolution was accompanied by the resolution of pneumonia.

Supplemental Information

Supplemental Information 1 Droplet digital PCR validation of microarray data.

Expression values obtained from the microarrays for IFNB1 and IL6 genes showed a significant positive correlation, confirmed by using digital droplet PCR.

Click here for additional data file.

Supplemental Information 2 Pulmonary gene expression profiles at day 5 and 10 post infection.

A) Venn diagram showing those genes whose expression levels differed from controls either at day 5 and day 10, and those which differed only at one time point. B) Heatmap of the common signature across different time points. The colour is proportional to their fold change (FC) compared to mock group, with the scale ranging from −4.2 FC (blue) to 4.2 FC (red).

Click here for additional data file.

Supplemental Information 3 List of genes differentially expressed between infected mice and controls.

FC: fold change. Highlighted genes in colour represent genes involved in the top 20 canonical pathways identified by IPA. Orange: represent cytokines or chemokines genes and blue represent interferon stimulated genes. Red represents granzyme molecules.

Click here for additional data file.

Supplemental Information 4 Genes involved in the top 20 canonical signaling pathways altered by A (H1N1) pdm 09 virus at day 5 post infection.

FC: fold change.

Click here for additional data file.

Supplemental Information 5 Variation of gene expression along time in the infected group.

Highlighted genes in colour represent genes involved in the top 20 canonical pathways identified by IPA. Orange: represent cytokines or chemokines genes and blue represent interferon stimulated genes. Red represents granzyme molecules.

Click here for additional data file.

Supplemental Information 6 Studies evaluating host transcriptomic responses to A (H1N1) pdm09 influenza virus in animal models.

This table summarized the most significant previous studies evaluating the transcriptomic response to A (H1N1) pdm09 virus in different animal models. EID50: 50% Egg Infective Dose, TCID50: 50% Tissue Culture Infective Dose, PFU: Plaque Forming Unit, dpi: days post infection.

Click here for additional data file.

Supplemental Information 7 Microarray raw data.

Click here for additional data file.

The authors kindly thank to the BSL3 facility staff for the technical support they provided during the experimental infection period. The authors are also grateful to Dr. Tomas Pumarola’s laboratory from Hospital Clinic of Barcelona in Spain, for the generous gift of human pandemic influenza A virus, A/Catalonia/63/2009.

Additional Information and Declarations

Competing Interests

Author Contributions

Animal Ethics

Microarray Data Deposition

Data Availability

The authors declare that they have no competing interests.

Raquel Almansa conceived and designed the experiments, performed the experiments, analyzed the data, contributed reagents/materials/analysis tools, wrote the paper, prepared figures and/or tables, reviewed drafts of the paper.

Pamela Martínez-Orellana conceived and designed the experiments, performed the experiments, analyzed the data, contributed reagents/materials/analysis tools, reviewed drafts of the paper.

Lucía Rico performed the experiments, reviewed drafts of the paper.

Verónica Iglesias performed the experiments, reviewed drafts of the paper.

Alicia Ortega performed the experiments, reviewed drafts of the paper.

Beatriz Vidaña performed the experiments, reviewed drafts of the paper.

Jorge Martínez performed the experiments, reviewed drafts of the paper.

Ana Expósito analyzed the data, reviewed drafts of the paper.

María Montoya conceived and designed the experiments, analyzed the data, contributed reagents/materials/analysis tools, reviewed drafts of the paper.

Jesús F. Bermejo-Martin conceived and designed the experiments, analyzed the data, contributed reagents/materials/analysis tools, wrote the paper, reviewed drafts of the paper.

The following information was supplied relating to ethical approvals (i.e., approving body and any reference numbers):

The ethical protocol and the research were reviewed and approved by the Animal and human Experimentation Ethical Committee of the Autonomous University of Barcelona (Internal Register Number: 1124M2R) and the Ethical Animal Experimentation Commission of the Catalan Government (Register Number: 5767).

The following information was supplied regarding the deposition of microarray data: E-MTAB-3866.

https://www.ebi.ac.uk/arrayexpress/experiments/E-MTAB-3866/

The following information was supplied regarding data availability:

Microarray expression data sets were uploaded at the Array Express microarray data repository and are already available publicly under accession number E-MTAB-3866.

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
