# Peer review of "Pulmonary transcriptomic responses indicate a dual role of inflammation in pneumonia development and viral clearance during 2009 pandemic influenza infection"

_PeerJ, doi:10.7717/peerj.3915_

## Round 0.1 · original submission · Major Revisions

Thank you again for submitting this manuscript to PeerJ. Please address the concerns of the reviewers to revise your manuscript.

Please also be careful to:

1. Discuss differences in your findings with Pommerenke et al., 2012 in more depth.
2. Discuss the microarray results from the 2 other time points that you currently do not discuss. Be careful to also discuss this in the context of already published work.

Reviewer 1 ·

Basic reporting

This is a descriptive study examining the transcriptome changes that occur during a mild H1N1 2009 influenza infection in mice. The manuscript is coherently written and laid out in a logical and cogent manner. However there are several weaknesses to the manuscript that reduce enthusiasm for acceptance.

1) Although the authors claim no research has been done to examine the non-fatal 2009 flu model, this is untrue. There have been multiple studies performed on H1N1 2009 (Itoh et al Nature 460, 1021-1025, Rowe et al 2010 Virology. Jun 5; 401(2): 257–265 ). to name a few. These should be referenced and more emphasis should be placed on what the purpose of this study is and how it differs from the other studies (CA2009 vs Cat2009 etc).

2) There needs to be more thorough editing performed on the paper, for instance:
line 51- "this severe" should be changed to "these severe"
line 56 "severe patients infected" should read "severely infected patients"
line 64- "previously put in relationship" would read better as "previously correlated with"

Experimental design

The experimental procedures seem well defined and the authors demonstrate a previously undescribed mouse influenza model. The numbers of mice seem appropriate and the statistical methods are fine. There is a little confusion in the materials and methods section that need to be clarified as well as some figures that need improvement.

1) Figure 2 demonstrates severe pathology at day 1 and 5. It is unclear what the differences are between day 1 and 5 though. This figure would greatly benefit from histopathological scoring. It is stated in the results section, but as the figure stands, it is showing very severe pathology at day 1 and not explaining that this was only from 2 of the mice.

2) it is confusing why they state "data not shown" on line 102 yet, they seem to have shown the data.
3) line 128 "the to" should be changed.
line 135: the authors say "as described elsewhere" that is fine, but give a citation showing where.
4) Line 176: Change "significantly higher body weight loss" to significantly greater weight loss"
5) line 182: "one day after mice infection" remove the word mice
6) line 183: Please use scientific notation for the standard deviation. same for line 184.
7) Line 225: change "induced also" to "also induced".

Validity of the findings

1) Since the 2009 H1N1 outbreak there has been extensive studies into this viral strain and the effects in multiple models (in vitro, mouse, ferret, pig) as well as in patients themselves. The discussion would greatly benefit from a better explanation of how this study ties in to the already abundant data that is out there.

2) The authors use a human isolate and it is unclear whether it has been passaged in mice already. The discussion would benefit with a comparison of well characterized fatal (PR8 or multiple mouse passaged 2009) infections.

3) It is unclear why the authors focused solely on day 5 when they have data from 3 different time points. The manuscript would benefit in a comparison of the changes from day 1, to day 5 to day 10.

·

Basic reporting

This article is reported in clear and unambiguous manner. There are some grammatical errors and I hope I have highlighted them all in my review.

There is an exhaustive list of references and an acknowledgement of an article (Pommerenke et al., 2012), which is quite similar in some aspects. The cited literature provides a good background and context to the study.

Figures and tables are provided in the main text and as supplementary data. There are clear and well structured. In the legend of Figure 1, statistics are presented in part B, but there is no asterisk on the viral load graph (B). I would recommend that statistics be presented out with the A and B description.

In figure 2 would be it possible to use arrows to indicate the infiltrate in the viral infected lungs

Figure 3 is a clear representation of differences in GEP, it would be beneficial to relate this figure to a table of genes that are a different within the figure legend. In this way it would be easier for reader to follow the process.

Does Table 1 represent only genes that are unregulated? could examples of down regulated genes also be included?

Experimental design

This study represents primary research within the aims and scope of the journal. The research question is clear, but I would record highlighting how it advances knowledge in this area and expand on lines 239 and 240 to explain what this study offers.

It is good to see microarray data accompanied by histological data. Methods and ethics are clearly described.

Validity of the findings

Data is robust and in line with previous literature.

Conclusion is well stated and linked to the research question.

Additional comments

This study is clear, easy to read and a good example of the importance of microarray data to illustrate a global view of immunological responses in a specific tissue.

I would recommend clarifying some of the figure legends to ensure that figures are fully explained in regards to the complexity of microarray data. For example in figure 3, I understand that the GEP profile is changing, but my immediate question is, what is changing? Can the figure 3 legend be connected to table 1 or supplementary table? Can table 1 be coloured to represent how it refers to figure 3 data.

---

## Round 0.2 · accepted · Accept

Thank you for taking into account the reviews and revising your manuscript accordingly.

Reviewer 1 ·

Basic reporting

The manuscript is vastly improved from the last version and the authors were very responsive to the critiques. The discussion would benefit from a more thorough editing for proper english.

Experimental design

the experimental design is well defined and meaningful.

Validity of the findings

the data is robust and well stated.